# Sniffin' Sticks and Olfactometer-Based Odor Thresholds for n-Butanol: Correspondence and Validity for Indoor Air Scenarios

**Marlene Pacharra [1,2,*], Stefan Kleinbeck [2], Michael Schäper [2], Christine I. Hucke [2] and Christoph van Thriel [2,*]** 

[1]  MSH Medical School Hamburg, University of Applied Sciences and Medical University, Am Kaiserkai 1, D-20457 Hamburg, Germany

[2]  Leibniz Research Centre for Working Environment and Human Factors at TU Dortmund University, Ardeystr. 67, D-44139 Dortmund, Germany; kleinbeck@ifado.de (S.K.); schaeper@ifado.de (M.S.); hucke@ifado.de (C.I.H.)

*   Correspondence: marlene.pacharra@medicalschool-hamburg.de (M.P.); thriel@ifado.de (C.v.T.); Tel.: +49-40-361-2264-9333 (M.P.); +49-231-1084-407 (C.v.T.)

**Abstract:** Threshold assessments for the reference odorant n-butanol are an integral part of various research, clinical, and environmental sensory testing procedures. However, the practical significance of a high or low threshold for n-butanol beyond a particular testing environment and procedure are often unclear. Therefore, this study aimed to determine between-method correlations and to investigate the association between the n-butanol threshold and perceptual/behavioral odor effects in natural breathing scenarios in 35 healthy adults. The thresholds for n-butanol derived from the Sniffin' Sticks test and determined by the ascending limit dynamic dilution olfactometry procedure were significantly correlated ($|r| = 0.47$). However, only the thresholds determined by olfactometry were significantly correlated to the odor detection of n-butanol in an exposure lab. Moreover, participants with a higher sensitivity for n-butanol in the olfactometer-based assessment rated ammonia, during a 75 min exposure, to be more unpleasant and showed better performance in a simultaneous 3-back task than participants with lower sensitivity. The results of this study suggest that beyond the strict parameters of a certain psychophysical procedure, the threshold for n-butanol can be a meaningful indicator of odor detection and effects in some cases.

**Keywords:** odor threshold; olfactometry; Sniffin' Sticks; chemosensory perception; validity assessment

## 1. Introduction

In clinical, research, and environmental assessment practice, odor sensitivity is currently determined almost exclusively with n-butanol (CAS: 71-36-3) as a reference odorant. As a consequence, parts of the clinical diagnosis of anosmia, the selection of panel members for sensory emission testing, and participation in olfactory research experiments can depend on an individual's threshold for n-butanol [1,2]. Moreover, n-butanol is one of the more abundant and relevant volatile organic compounds (VOCs) in indoor air environments. The German Environment Agency (UBA) mentioned in their indoor air guidance value document for 1-butanol (synonymical to n-butanol) that this VOC was found in 75–90% of indoor air samples in various databases and surveys [3]. Based on the developmental toxicity of 1-butanol, a health hazard guide value (RW II) of 2 mg/m$^3$ and a precautionary guide value (RW I) of 0.7 mg/m$^3$ were derived. The UBA report also stated that the RW I is above the odor threshold and that the olfactory perceptions need additional considerations.

Regardless of the relevance of n-butanol as an indoor air pollutant, empirical evidence is lacking as to whether sensitivity to n-butanol is an adequate marker for sensitivity to other odorants as well as for n-butanol itself outside of a given lab environment and testing procedure [4,5].

Odor delivery methods and psychophysical testing procedures used to derive the odor threshold for n-butanol vary widely between areas of application. This may give rise to a between-method variability in thresholds. While the Sniffin' Sticks test [6] is very common in research and clinical practice, dynamic dilution olfactometry is the most common method in environmental practice (see DIN EN 13725 [7]). The single staircase, 3-alternative forced choice procedure used in the Sniffin' Sticks test adapts every subsequent step to the individual's previous performance [6]. As this technique is difficult to implement when testing several participants simultaneously, dynamic olfactometry, as used during environmental odor evaluation procedures [8], relies on an ascending limit procedure [2].

While a recent report indicated a non-significant correlation between n-butanol thresholds determined with the Sniffin' Sticks test and ascending limits olfactometry ($r = 0.27$) [4], another study comparing sniff bottles and olfactometry methods for n-butanol and ammonia (CAS: 7664-41-7) reported adequate between-method correlations (e.g., $r = 0.78$) [9]. With regard to the real-life impact of n-butanol thresholds, there is some indication that a lower Sniffin' Sticks threshold for n-butanol is associated with lower pleasantness ratings for different odors presented in glass jars [10]. However, necessary parts of olfactometry and the Sniffin' Sticks tests are (a) prompted sniffing at a clearly identifiable odor source and/or (b) artificial breathing rhythms. Thus, the association between the odor thresholds derived from these methods and the odor detection and evaluation of environmental odors presented more naturally in the ambient air is so far unclear.

Given the practical importance of thresholds for n-butanol in clinical, research, and environmental assessment practice, the aims of the current study were threefold. Firstly, the between-method correlation (concurrent validity) was assessed for n-butanol thresholds determined with the very common Sniffin' Sticks test [6] and the established ascending limit dynamic dilution olfactometry procedure [2]. Secondly, the correspondence of these established threshold tests with the odor detection of n-butanol in indoor air scenarios was tested using an exposure lab. Thirdly, the association of these thresholds with odor effects caused by ammonia in an exposure lab was investigated. As the odors are presented in the ambient air, the exposure lab should more closely mimic the situation in the real world. Thus, the results of the here presented exposure lab experiments should be helpful in determining the ecological validity of the Sniffin' Sticks and olfactometry-based n-butanol thresholds.

To this end, a novel ascending limits procedure presenting a stair-wise increasing concentration of n-butanol under normal breathing conditions in an exposure lab was conducted, and its results correlated with the results of the established methods (Sniffin' Sticks and olfactometry). Moreover, the transferability of the results to the malodorous compound ammonia and its odor effects was tested; it was investigated whether the n-butanol thresholds derived using Sniffin' Sticks or olfactometry are associated with the perceptual and behavioral odor effects of the malodorous compound ammonia in a well-controlled natural breathing scenario simulated by means of an exposure lab experiment [11,12]. To compare the results of individuals more and less sensitive to n-butanol during ammonia exposure and, in this way, to mimic the potential behavior of different selected panelists in real-world scenarios, subgrouping of the sample was performed using cut-off values from a large normative sample (Sniffin' Sticks) [1] or the DIN EN 13725 norm (80 ppb) [7].

## 2. Experiments

### 2.1. Participants

Thirty-nine non-smoking participants were recruited for this experiment. Exclusion criteria included pregnancy, asthma, and acute or chronic upper airway diseases. Four participants were excluded from the data analysis to avoid unclear or biased odor thresholds; three participants had

increased false alarm rates during the olfactometer threshold test (>mean + 2 SD) (cf. [13]), and one participant indicated that he could not detect an odor at all in the exposure lab threshold test.

Thus, the final sample comprised 35 participants. For descriptive details, see Table 1. To evaluate if the number of subjects was sufficient, a power analysis (G-Power; [14]) was conducted. The expected correlations should be in the range of the test-retest reliabilities of the established olfactory detection threshold tests (e.g., for Sniffin' Sticks, between 0.43 and 0.85 [15]; 0.61 [6]; 0.92 [16]). Thus, for the comparison of different methods, we expected a correlation (Pearson *r*) of about 0.60 (see also [9], *r* = 0.78 correlation between sniff bottles and olfactometry). With 35 subjects, a statistical power of $1 - \beta = 0.97763$ could be achieved [17].

**Table 1.** Descriptive statistics for the total sample.

| Subject Characteristics | Total Sample |
|---|---|
| Men/Women (n) | 12/23 |
| Age (mean (SD)) | 23.8 (3.1) |
| CSS-SHR (mean (SEM)) | 31.8 (1.3) |
| Negative affectivity (mean (SEM)) | 14.0 (0.7) |
| FEV1 (mean (min-max)) | 96.4% (84.8–111.1%) |

Note: SD = standard deviation, SEM = standard error of the mean, CSS-SHR = Chemical Sensitivity Scale for Sensory Hyperreactivity, FEV1 = forced expiratory volume in 1 s.

### 2.2. Procedure

The ethics committee of the Leibniz Research Centre for Working Environment and Human Factors (IfADo) approved the study protocol (approval date: 23 March 2016), and written informed consent was obtained from all participants. The participants received no feedback about their test performance in any of the performed tests at any point during the study. They were instructed not to talk to the other participants about their odor perceptions during any of the tests or during the ammonia exposure. The study procedure is depicted in Figure 1.

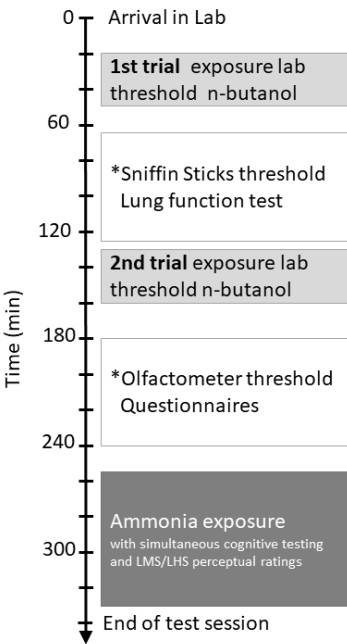

**Figure 1.** Study procedure. * blocks were switched randomly for half of the participants. LMS = labeled magnitude scale, LHS = labeled hedonic scale.

After arrival in the lab and giving informed consent, groups of 3–4 participants were administered the first trial of the n-butanol threshold procedure in the exposure lab. After completion, a 15 min break followed. Participants were assigned according to an a priori computed randomization scheme to one of two groups, which differed in the order the following detection tests were presented (see Figure 1): half of the participants (Group 1) first completed the olfactometer threshold assessment in groups of two participants and answered the Chemical Sensitivity Scale for Sensory Hyperreactivity [18] and the trait version of the Positive and Negative Affect Schedule [19]. The other half of the participants (Group 2) were first administered, individually, the Sniffin' Sticks threshold test and a lung function test (VitaloGraph, Hamburg, Germany).

In accordance with the GOLD guidelines [20] a forced expiratory volume in 1 s (FEV1) value $\leq 80\%$ in the lung function test was used as an indicator of asthma and chronic obstructive pulmonary disease. Accordingly, subjects with lower FEV1 values would have been excluded from the experimental exposure to ammonia. As only non-smoking, young, and healthy volunteers were enrolled, none of the participants had a FEV1 value below 80% (see Table 1) [20]. Then, all participants completed the second trial of the threshold procedure in the exposure lab. After a 15 min break, participants completed either the Sniffin' Sticks and the lung function test or the olfactometer test and questionnaires, depending on which tests they had already been administered by this point.

After a 15 min break, all participants underwent the 75 min ammonia exposure in the exposure lab. During ammonia exposure, cognitive testing, namely the n-back task [21] and flanker task [22], and perceptual ratings (via labeled magnitude scale, LMS; [23]) were conducted. The LMS is characterized by a quasi-logarithmic spacing of verbal labels and mimics the ratio-like properties of magnitude estimation scaling [23]. Furthermore, for hedonic scaling, the labeled hedonic scale was used (LHS; [24]) that is based on the LMS. The scale values for LHS and LMS in the computerized version used in this study ranged from 0 to 1000.

### 2.3. Materials

#### 2.3.1. Sniffin' Sticks-Based Threshold for n-Butanol

The Sniffin' Sticks (Burghart, Wedel, Germany) subtest for the assessment of the n-butanol threshold was used [1,6]. Here, the threshold value is defined as the average Sniffin' Stick number (lower numbers indication higher concentrations) of the last four reversals in a single-staircase, 3-alternative forced choice procedure.

Following the newest available norms of the test (see [25]), the cut-off score for individuals more and less sensitive to n-butanol using this test was 9 (median normative sample for age 21–30). Within this age range, there are only negligible differences between males and females, 8.75 for males and 9 for females (age 16–35; [1]), or more recently, 8.5 vs. 8.75 (age 21–30; [26]). As only non-smoking, healthy volunteers participated in the study, a cut-off value of 9 for males and females seemed to be appropriate.

#### 2.3.2. Olfactometer-Based Threshold for n-Butanol

A dynamic dilution olfactometer TO 8 (ECOMA GmbH, Kiel, Germany) was used that complies with DIN EN 13725 [2]. N-butanol was injected into 25 L Tedlar®-bags filled with nitrogen. The mixture was homogenized by heating and rotating the bag.

The standard procedure of the ascending method of limits with a 2-fold geometric dilution series was applied as in previous studies [13,27,28]. In short, the threshold measurement consisted of three trials in which increasing concentration steps of n-butanol were presented, interspersed with blank samples.

Participants had to press a button whenever they thought they detected an odor. The lower of two subsequent correctly identified concentration steps represented the estimate of reliable olfactory detection in that trial. The detection threshold was defined as the geometric mean of the three

trial estimates [13,27,28]. As in previous studies [13,28], the detection thresholds were subjected to log-transformations before data analysis.

According to DIN EN 13725 [7], a panel member for environmental odor testing should have an n-butanol threshold between 20 and 80 ppb [2]. There are no established, published thresholds that differentiate between males and females for the here used olfactometry test for n-butanol. Thus, the cut-off value for individuals more and less sensitive to n-butanol using the ascending limits olfactometry test was set to 80 ppb in this study.

### 2.3.3. Exposure Lab-Based Threshold for n-Butanol

The threshold assessment took place in a 28 $m^3$ exposure lab with four PC workstations. This environmental chamber has been used in previous experimental exposure studies, i.e., [29]. The assessment followed the same general procedure of the ascending method of limits as used for the olfactometer-based assessment [2]. Due to the higher time and operating costs of the exposure lab compared to the olfactometer, the assessment in the exposure lab consisted of only two instead of three trials.

In each trial, subjects were exposed over 30 min to an ascending concentration series of n-butanol (2-fold geometric series: 20, 40, 80, 160 and 320 ppb; see Supplement Figure S1). Every 5 min, subjects were prompted on a computer screen to indicate whether they detected an odor or not ("Odor? Yes/No"). Due to the technical restrictions in the lab, it was not feasible to insert randomly blank samples into the series. Thus, the first correctly identified concentration step represented the estimate of reliable olfactory detection in that trial. The detection threshold was defined as the geometric mean of the two trial estimates. Just as the olfactometer-based thresholds [13,28], the detection thresholds derived from the exposure lab procedure were subjected to log-transformations before data analysis.

### 2.3.4. Experimental Ammonia Exposure

The procedure as described in previous studies [11,12] was applied. In short, subjects were exposed to an ascending concentration of ammonia (CAS: 7664-41-7) over 75 min. The maximum concentration after 75 min was 10 ppm (see Supplement Figure S4) corresponding to 50% of the German maximum workplace concentration (MAK value) [30]. This concentration is clearly above previously published odor thresholds but still well below the lateralization thresholds [28]. To estimate the odor effects of ammonia during the exposure, chemosensory perceptions were rated via the LMS [23] and the LHS [24]. Further, cognitive performance was assessed using a 3-back working memory and response inhibition task (see Supplementary Figure S3).

### 2.3.5. Air Monitoring in the Exposure Lab

The 28 $m^3$ laboratory was supplied with conditioned air by a climate control unit in a neighboring room (temperature, 24.4 °C; humidity, 46.0%). A predefined amount of n-butanol or ammonia (experimentally determined by volumetric analysis) was mixed into the inlet airstream of the climate control system. The conditioned air was dispersed throughout the laboratory by a branched pipe system, which was located on the floor. The outlet system at the ceiling of the laboratory was actively controlled through four outlets by an exhaust air ventilator; it maintained the laboratory at a negative pressure of 20–30 Pa. The air exchange rate was approximately 300 $m^3$/h.

Air samples were taken from the airflow of the inlet pipe and from the inside of the exposure laboratory quasi-continuously (every 80 s) during all exposure sessions. Photo acoustic IR spectroscopy was used to analyze the air samples (INNOVA, 1412i Photo Acoustic Field Gas-Monitor, LumaSense, Ballerup, Denmark). An overview of measured concentration values for n-butanol and ammonia is given in the Supplement (Supplementary Figures S2 and S4).

*2.4. Statistical Analysis*

The statistical analyses were performed in IBM SPSS Statistics 24. The level of significance for all statistical tests was set to 0.05. We checked for outliers by using the more liberal definition of extreme outliers ("outer fences": Q3 + 3 × IQR) [31], and according to this criterion, all participants could be included in the analysis.

Based on the two thresholds, participants were classified into a 2 × 2 table below or above the respective cut-off values. Pearson's chi-square and exact tests were used to analyze the association of the grouping results. Moreover, the group differences for the Sniffin' Sticks scores and the olfactometer-based threshold were analyzed by Mann–Whitney U tests.

A Pearson correlation was computed between the Sniffin' Sticks-based and olfactometer-based threshold for n-butanol to compare the methods.

Next, the two established thresholds were correlated with the exposure lab-based threshold using further Pearson correlations. All correlations were adjusted (Bonferroni method) for the total number of computed multiple comparisons. Bonferroni-adjusted *p*-values are shown in addition to the non-adjusted correlations for these analyses.

The experimental data from the ammonia exposure were analyzed using full-factorial analyses of variance (ANOVAs), with time as the repeated measures factor and group as the between-subjects factor. Models were calculated taking into account, on the one hand, the grouping factor Sniffin' Sticks threshold (cut-off value: 9, see Sniffin' Sticks norms) and taking into account, on the other hand, the grouping factor olfactometer-based threshold (cut-off: 80 ppb, see DIN EN norm 13725 [7]). If the assumption of sphericity was violated, Greenhouse–Geisser-corrected degrees of freedom were used. Significant interaction effects were further analyzed using Bonferroni-adjusted post hoc tests.

## 3. Results

*3.1. Results of the Psychometric Threshold Assessments*

Table 2 presents the descriptive statistics of the three olfactory measures of n-butanol sensitivity for the total sample and after applying the respective cut-offs. Unsurprisingly, when a cut-off was applied based on one of the thresholds, Mann-Whitney U tests indicated a significant difference between resultant groups in this threshold. Moreover, participants more and less sensitive in the Sniffin' Sticks tests also differed significantly in their olfactometry-based threshold. Participants did not differ in relevant psychological variables for odor effects [29] such as negative affectivity and self-reported chemical sensitivity (see supplement Table S1).

**Table 2.** Description of total sample and classified subgroups.

| Subject Characteristics | Total Sample | Sniffin' Sticks Threshold | | Olfactometer Threshold | |
|---|---|---|---|---|---|
| | | <9 | ≥9 | >80 ppb | ≤80 ppb |
| Men/Women (n) | 12/23 | 7/14 | 5/9 | 6/10 | 6/13 |
| Sniffin' Sticks T No. pen (median (IQR)) | 8.0 (6.5–9.8) | 6.8 (6.3–8.0) | 9.8 * (9.3–10.8) | 8.0 (6.5–9.1) | 8.3 (7.3–10.8) |
| Olfactometer T ppb (median (IQR)) | 80 (50–160) | 101 (64–160) | 45.2 * (32–127) | 160 (127–228) | 50.4 * (32–80) |
| Exposure lab T ppb (median (IQR)) | 80 (40–113) | 80 (57–113) | 68.3 (40–113) | 136.6 (48–226) | 80 (40–113) |

Note. IQR = inter-quartile range, T = threshold, * *p* ≤ 0.05 subgroup comparison using Mann-Whitney U tests.

### 3.2. Results of the between-Method Correlations for n-Butanol Thresholds

The correlation between the Sniffin' Sticks- and olfactometer-based threshold (see Figure 2) was significant ($r = -0.47$; $p = 0.004$, Bonferroni-adjusted $p = 0.012$).

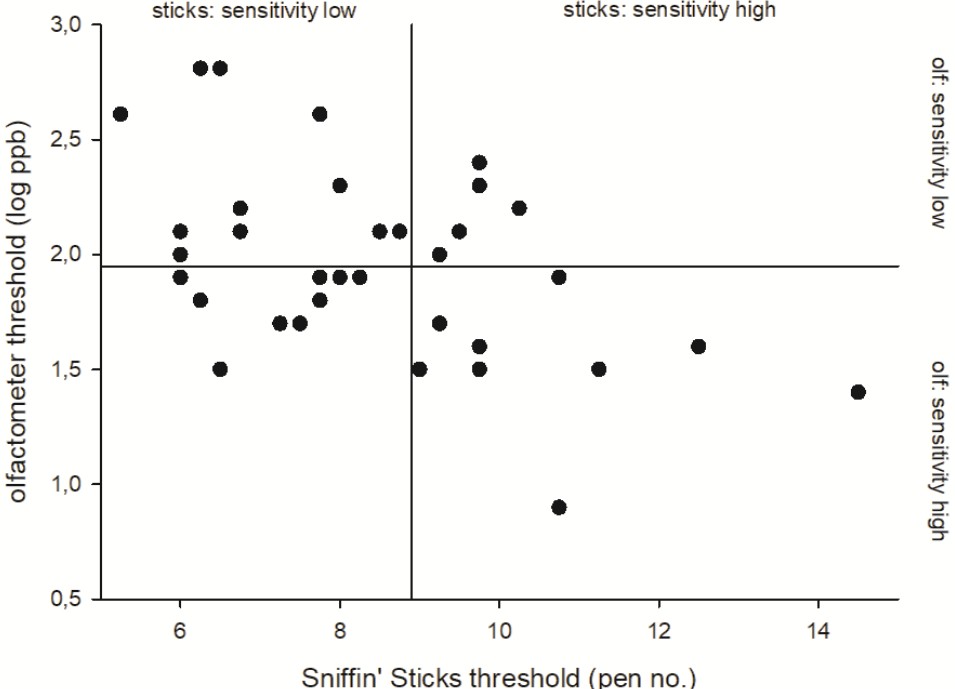

**Figure 2.** Scatter plot depicting the association between n-butanol Sniffin' Sticks- and olfactometer-derived thresholds. Note that in the Sniffin' Sticks test, a higher pen number corresponds to a higher n-butanol dilution and thus a lower threshold (higher sensitivity). Vertical and horizontal lines depict the respective cut-off values for high vs. low sensitivity groups (for details see text).

When applying the cut-off values for the Sniffin' Sticks- ($\geq$9) and olfactometer-based ($\leq$1.9 log ppb) thresholds, nine participants (25.7%; lower right quadrant) were classified as individuals with high olfactory sensitivity in both standardized n-butanol threshold assessments (cf. Figure 2). Three of these participants were males (three out of 12; 25%) and the other six were females (six out of 23; 26%). However, the statistical analysis of the 2 × 2 contingency table yielded a non-significant Pearson chi-square value of 0.94 ($p = 0.49$). Thus, there was no significant overlap of the two olfactory sensitivity classification approaches.

Both thresholds for n-butanol (olfactometer and Sniffin' Sticks) were correlated with the exposure lab-based threshold for n-butanol (Figure 3). Due to repeated computation of correlations with the same participants, a Bonferroni adjustment of $p$-values was conducted, resulting in a significant correlation between the olfactometer-based threshold and the exposure-lab based threshold ($r = 0.41$, $p = 0.015$, Bonferroni-adjusted $p = 0.045$, see Figure 3a). However, the Bonferroni-adjusted correlation between the Sniffin' Sticks-based threshold and the exposure lab-based threshold was non-significant ($r = -0.34$, $p = 0.048$, Bonferroni-adjusted $p = 0.144$, see Figure 3b).

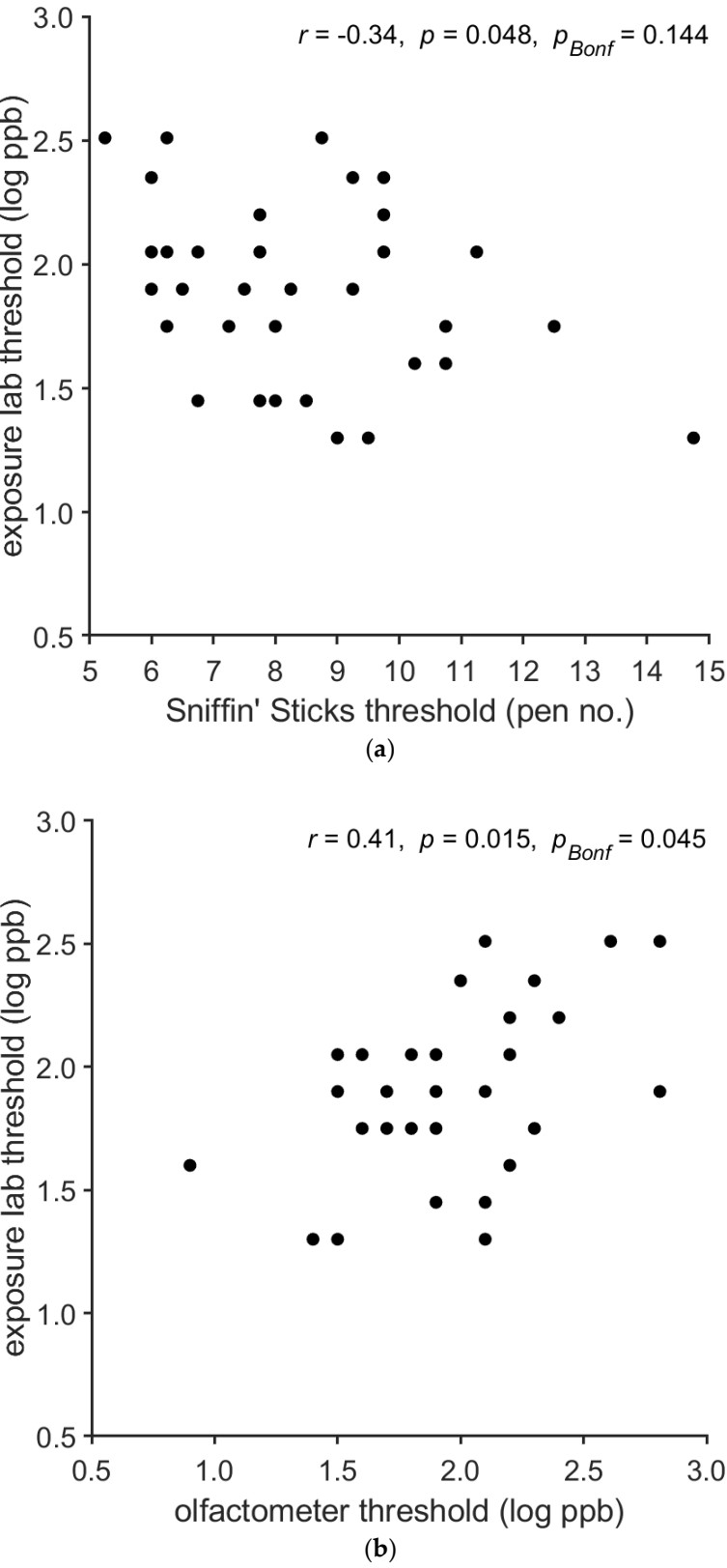

**Figure 3.** Scatter plots depicting the associations between the n-butanol thresholds derived with the exposure lab and the investigated methods, (**a**) Sniffin' Sticks and (**b**) olfactometer. Note that in the Sniffin' Sticks test, a higher pen number corresponds to a higher n-butanol dilution and thus a lower threshold (higher sensitivity).

In a second step, it was investigated whether subgrouping the participants into more and less sensitive individuals based on detection thresholds was associated with odor effects for the compound ammonia. As only the olfactometry-derived thresholds showed a significant association with the exposure lab n-butanol detection threshold, a cut-off score of 80 ppb in the olfactometer-based assessment was used in the following analyses to indicate individuals more and less sensitive to n-butanol.

### 3.3. Results of the Modulation of Odor Effects by n-Butanol Thresholds

### 3.3.1. Chemosensory Perceptions

As expected, perceptual ratings were affected by the concentration of ammonia; participants perceived ammonia to be more unpleasant, intense, and pungent with increasing concentration (all main effects of concentration, $p < 0.001$; see Figure 4).

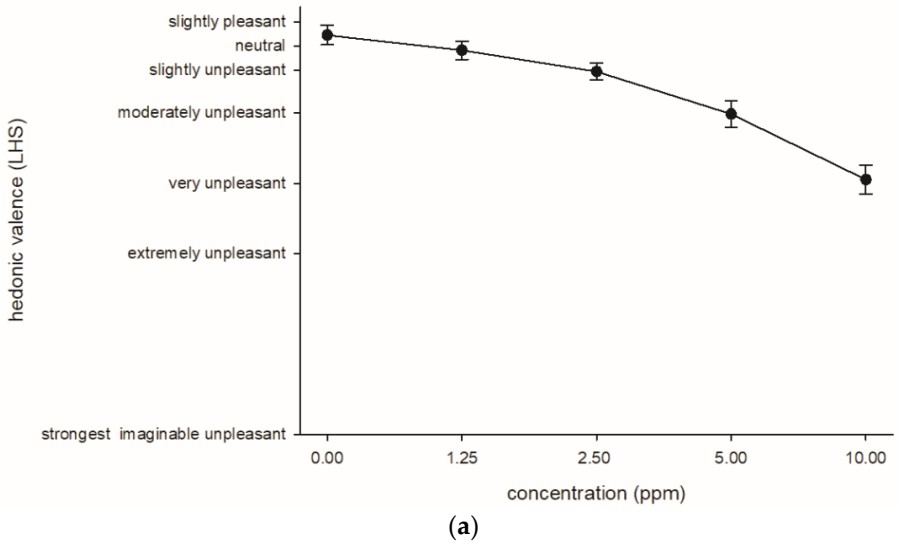

(a)

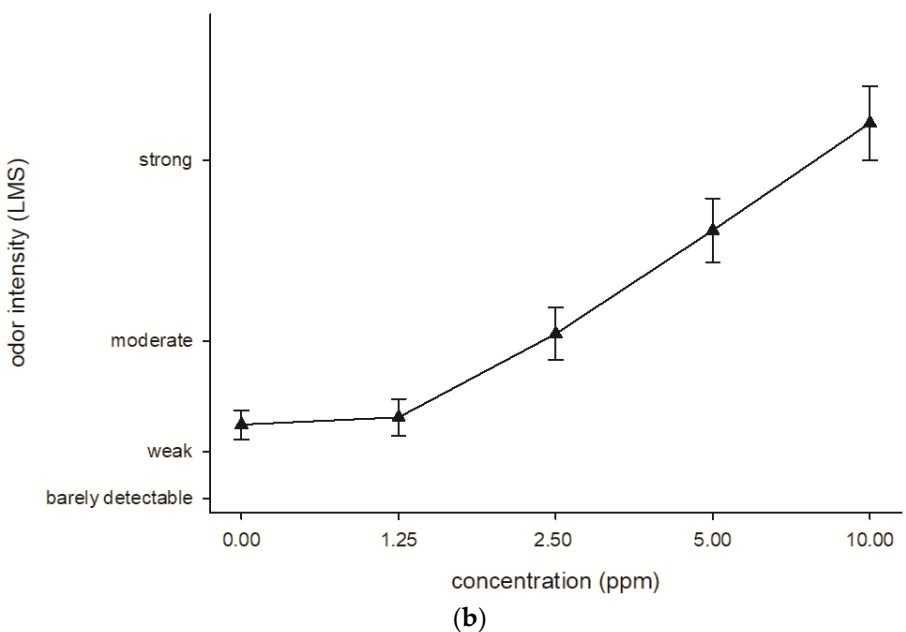

(b)

**Figure 4.** *Cont.*

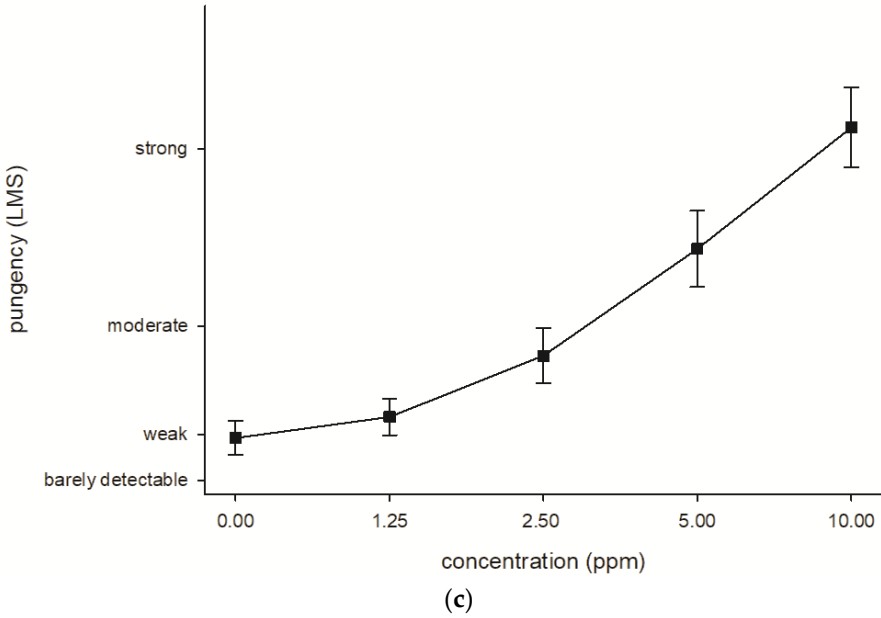

(**c**)

**Figure 4.** Impact of different concentrations of ammonia on perceived (**a**) hedonic value, (**b**) odor intensity and (**c**) pungency (mean ± SEM).

A significant main effect of the olfactometer-based threshold on pleasantness ratings emerged, $F(1,33) = 4.2$, $p = 0.049$. Participants with a lower olfactometer-based threshold (higher sensitivity) rated the exposure to be more unpleasant (mean = 426, SEM = 12; scale range: 0–1000) than participants with a higher olfactometer-based threshold (mean = 463, SEM = 13) (see Figure 5).

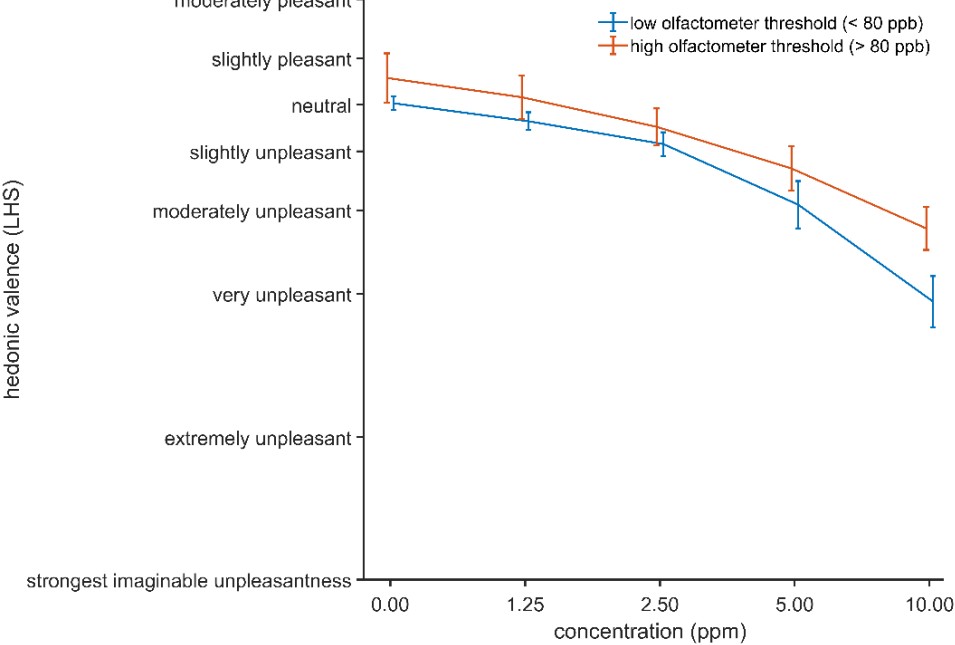

**Figure 5.** Effect of olfactory sensitivity assessed via the olfactometer-based threshold on pleasantness ratings during ammonia exposure (mean ± SEM).

Figure 5 indicates that the difference between the two groups increased with increasing ammonia concentration. However, the interaction of the sensitivity group and concentrations was not significant.

### 3.3.2. Odor Effects on Behavioral Task Performance

Participants improved their performance in the 3-back and response inhibition tasks over the course of the test session as indicated by an increase in the percentage of correct responses and a decrease in reaction times (all main effects of concentration, $p \leq 0.05$).

With regard to the olfactometer-based threshold for n-butanol, significant main effects on reaction times, $F(1,33) = 19.7$, $p < 0.001$, and error rates, $F(1,33) = 5.4$, $p = 0.026$, in the 3-back task emerged. Participants with a lower olfactometer-based threshold (higher sensitivity) had shorter reaction times and a higher percentage of correct responses in the 3-back task compared to participants with a higher olfactometer-based threshold (see Figure 6).

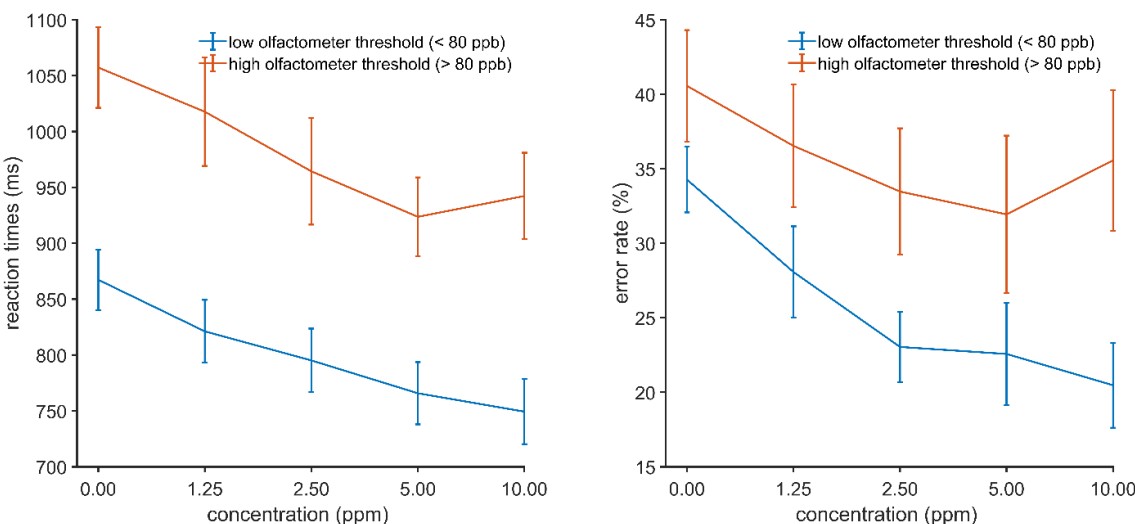

**Figure 6.** Effect of olfactometer-based threshold on 3-back reaction times (**left**) and error rates (**right**) during ammonia exposure (mean ± SEM).

Comparable to the rating data (see Figure 5), there was no interaction with the increasing ammonia concentration, indicating no additional impact of the increasing chemosensory perceptions.

## 4. Discussion

Given the importance of n-butanol odor thresholds in many research, clinical, and environmental testing contexts, information on the practical significance of this particular odor threshold beyond the particular testing environment and procedure is scarce. This study sought to remedy that.

In contrast to a previous report [4], a medium-sized, significant correlation between the thresholds derived from the Sniffin' Sticks test and the ascending limit dynamic dilution olfactometry procedure could be shown. This indicates that the determined sensitivity to n-butanol is associated between these two established methods of threshold assessment [2,6] and supports the good between-method correlations (concurrent validity) previously reported for other threshold assessment methods [9,28].

Beyond established threshold procedures, a novel exposure lab-based threshold assessment for n-butanol was proposed that more closely mimics odor detection during natural breathing. Measured concentration values showed that an ascending concentration series similar to the olfactometer-based method [2] could be generated in an exposure lab. After Bonferroni correction, only a significant medium-sized correlation between n-butanol thresholds derived using olfactometry and this novel method emerged. This indicates that olfactometry-derived thresholds can be meaningful indicators of odor detection in a more realistic context. While the Sniffin' Sticks threshold test requires artificial breathing (e.g., sniffing), the olfactometry and exposure lab scenarios have in common that they allow a more natural breathing pattern.

Moreover, the results showed that a lower olfactometer-based threshold for n-butanol is associated with lower pleasantness ratings for ammonia during an exposure lab scenario. This further highlights the external validity of n-butanol thresholds with regard to perceptual effects during natural breathing of another odor and irritant (ammonia). Additionally, it is in line with a previous experimental finding [10], showing that the threshold for n-butanol is associated with lower pleasantness ratings for a range of odors presented in glass jars.

An interesting, secondary finding in this study constitutes the better cognitive working memory performance in those with lower olfactometer-based thresholds irrespective of the ambient ammonia concentration. With regard to the Sniffin' Sticks threshold for n-butanol, Hedner, et al. [32] reported that the threshold is unrelated to cognitive factors such as executive functioning, semantic memory, and episodic memory. However, whether this is also the case for olfactometer-based thresholds is so far unclear. As the two "high odor sensitivity" groups showed only a weak overlap (25.7%), factors unrelated to olfaction but relevant for cognitive task performance (e.g., education and IQ) might have caused this general performance difference.

In recent studies using gas chromatography-olfactometry, a coupling of gas chromatography analysis and human olfaction by panelists was employed to identify single VOCs in mixtures [33]. For n-butanol, a linear relationship was found between the modified detection frequency (frequency of detection × evaluation of intensity) of panelists and concentration of n-butanol as measured by gas chromatography (MS) in adhesives [34].

When humans inhale, ambient air is analyzed when reaching the olfactory epithelium. There, trace components of the air interact with receptor cells [35]. Thresholds and atmospheric lifetime are related in such a way that highly reactive odorants (short-lived molecules) are detected more sensitively [35]. N-butanol, belonging to the family of alcohols, therefore, has a relatively low odor threshold.

All threshold assessments in this study indicated that the median olfactory threshold for n-butanol in the experimental sample was higher than what would be expected from norm values [1] or permissible for panel members during sensory emission testing according to DIN EN 13725 [7] (compare Table 2). This would suggest an overall lower than average sensitivity to n-butanol in the sample. This could be due to (1) a sampling error associated with the low sample size, (2) undetected nasal obstruction in the participants, or (3) olfactory adaptation due to multiple assessments of the odor threshold for n-butanol on the same day.

Despite these possible confounding factors, the results showed that the threshold for n-butanol can be a meaningful indicator of odor detection and odor effects in natural breathing scenarios. This could be seen as a first step in providing much needed confidence in these thresholds [4,5] that are used daily in so many research and other application areas.

## 5. Limitations of the Study

Before coming to the conclusions, some limitations should be mentioned that need to be addressed in further studies. First, the sample size was sufficient to detect the association between n-butanol odor thresholds and the odor effects of another compound, but the sample was highly selective, and therefore, the transferability to the general population is somewhat limited. Here, a larger sample including older subjects, subjects with mild diseases of the upper respiratory tract (e.g., allergic rhinitis), and subjects reporting an increased odor sensitivity should be investigated. Second, the new method of the exposure lab-based threshold assessment should be tested with other odorants and compared to other threshold assessment procedures like squeezing and sniffing bottles [36–38] or the triangle bag method [39]. Third, odorants other and more pleasant than ammonia should be used to include the highly relevant dimension of pleasantness [40] into this branch of odor research.

## 6. Conclusions

The results presented here provide further empirical evidence that the olfactory sensitivity of an individual may be an important predictor of odor perceptions in near to realistic scenarios of the human

odor experience. The reference compound n-butanol seems to be an adequate choice as shown by the good cross-method correlations. Nevertheless, the role of suprathreshold olfactory functioning such as odor discrimination or identification has not been conclusively studied in this context. Moreover, other reference compounds for panelist selection are currently under discussion (DIN EN 13725:2019) [41]. With respect to the impact of environmental odors on cognitive task performance, our results showed that "high odor sensitivity" was not associated with worse performance in a challenging working memory task. The results were opposite to a distractive effect of malodors as proposed previously [42].

**Supplementary Materials:** The following are available online at http://www.mdpi.com/2073-4433/11/5/472/s1, Figure S1: Schematic overview of the experimental procedure during the exposure lab-based threshold assessment. Figure S2: Measured concentration values for n-butanol during the exposure lab-based threshold assessment. Figure S3: Schematic overview of the experimental procedure during the ammonia exposure (cf. [11,12]). Figure S4: Measured concentration values of ammonia during the experimental exposure. Table S1: Descriptive statistics for the total sample and subgroups.

**Author Contributions:** Conceptualization, M.P. and C.v.T.; methodology, M.P. and M.S.; validation, S.K. and M.S.; formal analysis, M.P. and S.K.; investigation, M.P. and C.v.T.; resources, C.v.T.; data curation, M.S.; writing—original draft preparation, M.P. and S.K.; writing—review and editing, M.P., S.K., M.S., C.I.H. and C.v.T.; visualization, M.P., C.I.H. and S.K.; supervision, C.v.T. All authors have read and agreed to the published version of the manuscript.

**Funding:** This research received no external funding.

**Acknowledgments:** The authors would like to thank Meinolf Blaszkewicz, Nicola Koschmieder, Eva Strzelec, Michael Porta, and Beate Aust for technical assistance. The publication of this article was funded by the Open Access Fund of the Leibniz Association.

**Conflicts of Interest:** Marlene Pacharra, Stefan Kleinbeck, Michael Schäper, Christine I. Hucke and Christoph van Thriel declare that they have no conflict of interest.

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
