# Peer review of "Sniffin’ Sticks and Olfactometer-Based Odor Thresholds for n-Butanol: Correspondence and Validity for Indoor Air Scenarios"

_atmosphere, doi:10.3390/atmos11050472_

Round 1

Reviewer 1 Report

The study of Pacharra et al. “Sniffin’ Sticks and olfactometer-based odor 2 thresholds for n-butanol: Correspondence and 3 validity for indoor air scenarios” aims to compare the different techniques used to establish the n-butanol threshold and to investigate the association between n-butanol threshold and perceptual/behavioral odor effects in natural breathing. I think the topic is very interesting and suitable for the issues of the Atmosphere journal.

However, I believe that the results obtained by the authors are not strong enough to support their conclusions, but this is not mentioned in the text. I suggest taking into account that some correlations are very weak and discussing it appropriately.

L77: I find that the sample size of this study is too small to even subdivide it into subgroups

Table 1: it is not clear to the reader what these numbers indicate 12/23 7/14 5/9 6/10 6/13. Again, standard error or standard deviation should be added where necessary to understand the sample variability

L 98: more information on the LMS scale used should be provided

L 186: this cut-off value does not take into account age and gender

L190: perhaps we should also test for the assumption of homogeneity of the variance

Figure 2: this data (p = 0.48) may not be considered significant

L 198: it would be advisable that the tags indicating the subjects appear in the plots, in order to be able to compare the results

L 201-203: the usefulness of this result is not clear. Was the sensitivity significantly different from the others? what are the 9 participants in the plots? I would have expected a separate cloud

Figura 3: the graphs are illegible, the writing should be larger

L 213-214: this result is not significant

L 221: authors should report the numbers

L 223-225: I suggest that the authors report the individual p value for each analysis, so that the readers can see the real values. This is because the conclusions of the study are based on the statistical results

L 249: this correlation is not supported by the data

L 252-253: given the poor quality of the figures presented, it is difficult for me to evaluate this statement

L 255: recent studies have also shown that the threshold for n-butanol is correlated with the ability of the subjects to smell single molecules during GC-O analysis

Author Response

The study of Pacharra et al. “Sniffin’ Sticks and olfactometer-based odor 2 thresholds for n-butanol: Correspondence and 3 validity for indoor air scenarios” aims to compare the different techniques used to establish the n-butanol threshold and to investigate the association between n-butanol threshold and perceptual/behavioral odor effects in natural breathing. I think the topic is very interesting and suitable for the issues of the Atmosphere journal.

We would like to thank the reviewer for the positive evaluation of our manuscript as well as for the detailed comments and suggestions that allow us to increase the quality of the revised version. Please find below the reply to the more specific comments.

However, I believe that the results obtained by the authors are not strong enough to support their conclusions, but this is not mentioned in the text. I suggest taking into account that some correlations are very weak and discussing it appropriately.

Reply: For the threshold correlations we adjusted the p-values following Bonferroni. The correlation between Sniffin’ Sticks threshold and exposure lab threshold does not reach significance anymore. The discussion is changed accordingly.

L77: I find that the sample size of this study is too small to even subdivide it into subgroups

Reply: A power analysis (G-Power) revealed that the sample size is high enough to find a correlation of .60 which resembles the mean test-retest reliabilities for Sniffin’ Sticks test found in the literature with a power of 1-β = .97763. Usually, a power of 1-β between .8 and .9 is recommended. The subdivision is done by a post-hoc dichotomization of olfactometer-based threshold for n-butanol. We abstained from doing a subdivision based on Sniffin’ Sticks-based threshold due to insignificant correlation to the exposure lab threshold for n-butanol in our analyses.

Table 1: it is not clear to the reader what these numbers indicate 12/23 7/14 5/9 6/10 6/13. Again, standard error or standard deviation should be added where necessary to understand the sample variability

Reply: Table 1 is simplified by now, showing only the descriptive statistics for the total sample. The number 12/23, therefore, indicates the ratio of males and females as indicated by the label (Men/Women).
The threshold values were now moved to another table (table 2: Description of classified subgroups) at the beginning of the results section (3. Results; 3.1. Results of the psychometric threshold assessments). This table is divided in 4 subgroups of which 2 comprise the whole sample each (whole sample divided by Sniffin’ Stick threshold dichomitization and whole sample divided by Olfactometer threshold dichomitization).

A combined 2x2 partition of both dichotomizations is applied to the figure showing the correlation between Sniffin’ Sticks threshold and Olfactometer threshold (figure 2). The number of subjects in each partition is described in the text beneath. With a non-parametric, test an asymmetric allocation to groups was excluded.

L 98: more information on the LMS scale used should be provided

Reply: L 135 “… and perceptual ratings [via labeled magnitude scale, LMS; 21] were conducted. The LMS is characterized by a quasi-logarithmic spacing of verbal labels and mimics the ratio-like properties of magnitude estimation scaling [21]. Furthermore, for hedonic scaling, the labeled hedonic scale was used [LHS; 22] that is based on the LMS. Scale values for LHS and LMS in the computerized version used in this study ranged from 0 to 1000.”

L 186: this cut-off value does not take into account age and gender

Reply: Age is considered as it is the median for this age group. Gender does not affect this cut-off value:

“… the cut-off score for individuals more and less sensitive to n-butanol using this test was 9 (median normative sample for age 21-30). Within this age range, there are only negligible differences between males and females: 8.75 for males and 9 for females [age 16-35; 1] or more recently 8.5 vs. 8.75 [age 21-30; 23]. As only non-smoking, healthy volunteers participated in the study a cut-off value of 9 for males and females seems to be appropriate.”

L190: perhaps we should also test for the assumption of homogeneity of the variance

Reply: The violation of sphericity occurs when it is not the case that the variances of the differences between all combinations of the conditions are equal. Thus, the assumption of homogeneity of variances has been tested and standardized statistical approaches (Greenhouse-Geisser-corrected degrees of freedom) were used to account for possible violations.

Figure 2: this data (p = 0.48) may not be considered significant

Reply: As we did a Bonferroni adjustment of the p-values for the correlations p = 0.048 is not considered significant anymore (p = .144).

L 198: it would be advisable that the tags indicating the subjects appear in the plots, in order to be able to compare the results

L 201-203: the usefulness of this result is not clear. Was the sensitivity significantly different from the others? what are the 9 participants in the plots? I would have expected a separate cloud

Figure 2 is now limited to correlation between Sniffin’ Sticks thresholds and Olfactometer thresholds. Lines in the plot now indicate the cut-off values forming the different subgroups. Details concerning the subgroups are now mentioned in the text:

“When applying the cut-off values for the Sniffin’ Sticks (≥ 9) and the olfactometer-based threshold (≤ 1.9 log ppb) nine participants (25.7 %; lower right quadrant) were classified as individuals with high olfactory sensitivity in both standardized n-butanol threshold assessments (cf. Figure 2). Three of these participants were males (3 out of 12: 25 %) and the other six were females (6 out of 23: 26 %). However, the statistical analysis of 2×2 contingency table yielded a non-significant Pearson Chi-Square value of 0.94 (p = 0.49). Thus, there was no significant overlap of the two olfactory sensitivity classification approaches.”

Another figure is added to the manuscript (Figure 3) showing the scatter plots of exposure lab thresholds with Sniffin’ sticks thresholds (a) and Olfactometer thresholds (b), respectively. Both scatter show also the Bonferroni-adjusted p-values.

Figura 3: the graphs are illegible, the writing should be larger

Reply: Figure 3 (meanwhile figure 4) is now larger and should be clearer

L 213-214: this result is not significant

Reply: As we stated in the statistical analysis section (2.4 Statistical analysis), the level of significance was set to 0.05. There is large agreement in the scientific community for this statistical significance level in two-sided tests. We do not see the reason why p = 0.049 should not be considered significant.

L 221: authors should report the numbers

Reply: Numbers are reported significant main effects of the olfactometer-based threshold on pleasantness ratings:
“… Participants with a lower olfactometer-based threshold (higher sensitivity) rated the exposure to be more unpleasant (mean = 426, SEM = 12; scale range: 0-1000) than participants with a higher olfactometer-based threshold (mean = 463, SEM = 13) (see Figure 4).”

L 223-225: I suggest that the authors report the individual p value for each analysis, so that the readers can see the real values. This is because the conclusions of the study are based on the statistical results

Reply: We report p-values for every significant effect. Usually, non-significant p-values were not reported. In some cases, we report p-values even for non-significant effects to emphasize that the p-value is far from significance. We think, reporting every p-value would be confusing.

L 249: this correlation is not supported by the data

Reply: As we did a Bonferroni adjustment, this correlation is not considered significant any more. The discussion has been changed accordingly.

L 252-253: given the poor quality of the figures presented, it is difficult for me to evaluate this statement

This result is not based on a figure but on statistical analyses. However, the figures are now bigger and should be clearer.

L 255: recent studies have also shown that the threshold for n-butanol is correlated with the ability of the subjects to smell single molecules during GC-O analysis

The following paragraph was added to the discussion:

“… In recent studies using gas chromatography-olfactometry using a coupling of gas chromatography analysis and human olfaction by panelists to identify single volatile organic compounds (VOCs) in mixtures [30]. For n-butanol, a linear relationship was found between modified detection frequency (frequency of detection x evaluation of intensity) of panelists and concentration of n-butanol as measured by gas chromatography (MS) in adhesives [31].  

When humans inhale, ambient air is analyzed when reaching the olfactory epithelium. There, trace components of the air interact with receptor cells [32]. Thresholds and atmospheric lifetime are related in a form that highly reactive odorants (short-lived molecules) are detected more sensitively [32]. N-butanol, belonging to the family of alcohols, therefore, has a low odor threshold. “

Reviewer 2 Report

This study explores the sensitivity of n-butanol by different olfactory assessment methods, and this topic is very attractive. In the study, n-butanol was used as the reference odor to determine the odor sensitivity, and ammonia was used as an environmental impact factor.

However, the messy presentation of data and the lack of a structural  analysis are difficult to understand.Figure 2 shows that the n-butanol threshold of Sniffin ’Sticks is independent of the other two analyses, thus the outliers should be evaluated separately. In Figure 3, the illustration is too small and unclear, showing the same dynamic linear relationship. I suggest that the current format of this paper needs to be substantially revised and restructured for discussion.

Author Response

This study explores the sensitivity of n-butanol by different olfactory assessment methods, and this topic is very attractive. In the study, n-butanol was used as the reference odor to determine the odor sensitivity, and ammonia was used as an environmental impact factor.

However, the messy presentation of data and the lack of a structural analysis are difficult to understand. Figure 2 shows that the n-butanol threshold of Sniffin’ Sticks is independent of the other two analyses, thus the outliers should be evaluated separately. In Figure 3, the illustration is too small and unclear, showing the same dynamic linear relationship. I suggest that the current format of this paper needs to be substantially revised and restructured for discussion.

We would like to thank the reviewer for the clear and critical evaluation of our experiments and the submitted manuscript. We think the comments were very helpful to reorganize the data presentation, to revise the statistical analyses, and to improve the quality of the manuscript.

Reply: The presentation of the data in all tables and figures has been adjusted and a clear structure of the analysis is now outlined already in the introduction (starting from line 73). Subheadings related to these aims were included into the result section. Figure 2 (old version) showed that the Sniffin’ Sticks scores were associated/ correlated with the two other threshold assessments and not independent as suggested by reviewer #2. However, due to the recommend Bonferroni correction the p-value of the correlation to the exposure lab-based n-butanol threshold was no longer considered significant in the revised version (see line 464).
If we understood the comment correctly the inclusion of the two “outliers” having a Sniffin’ Stick score above 14 or an olfactometer-based threshold below 1,0 log ppb should be addressed. As such values are realistic in odor threshold assessments (see normative sample of the Sniffin’ Sticks) we applied a more liberal statistical criterion for the detection of outliers (see line 219: “We checked for outliers by using the more liberal definition of extreme outliers (“outer fences”: Q3 + 3×IQR) [NIST/SEMATECH e-Handbook of Statistical Methods, https://www.itl.nist.gov/div898/handbook/prc/section1/prc16.htm] and according to this criterion all participant could be included in the analysis.”).
All figures were revised and rescaled and thereby, the readability should have been increased.

Reviewer 3 Report

The authors evaluated 39 subjects in order to compare three different methods for the measurement of n-butanol olfactory threshold, namely Sniffin’ sticks, dynamic diluition olfactometry and a new method based on exposure lab. Furthermore, the volunteers were after exposed to ammonia to assess whether there was a difference in assessment of its unpleasantness between those highly sensitive to n-butanol and those less sensitive to n-butanol. The authors found a significant correlation between the methods and a significant higher sensitivity to the unplesant odour in those more senditive to n-butanol.

Unfortunatelly, the number of patients considered (only 35) is very low, especially if the authors want to compare 3 different methods and one of these is new. At least a power test should be performed in order to confirm the validity of the statistical results.

The authors switched the test blocks for half the partecipants. Was it done after a computerized randomization? Please explain.

Author Response

The authors evaluated 39 subjects in order to compare three different methods for the measurement of n-butanol olfactory threshold, namely Sniffin’ sticks, dynamic diluition olfactometry and a new method based on exposure lab. Furthermore, the volunteers were after exposed to ammonia to assess whether there was a difference in assessment of its unpleasantness between those highly sensitive to n-butanol and those less sensitive to n-butanol. The authors found a significant correlation between the methods and a significant higher sensitivity to the unplesant odour in those more senditive to n-butanol.

Unfortunatelly, the number of patients considered (only 35) is very low, especially if the authors want to compare 3 different methods and one of these is new. At least a power test should be performed in order to confirm the validity of the statistical results.

The authors switched the test blocks for half the partecipants. Was it done after a computerized randomization? Please explain.

We would like to thank the reviewer for the comprehensive summary of our manuscript as well as for the greatly appreciated comments.

Reply: A power analysis (G-Power) revealed that the sample size is high enough to find a correlation of .60 which resembles the mean test-retest reliabilities for Sniffin’ Sticks test found in the literature with a power of 1-β = .97763. Usually, a power of 1-β between .8 and .9 is recommended. The subdivision is done by a post-hoc dichotomization of olfactometer-based threshold for n-butanol. We abstained from doing a subdivision based on Sniffin’ Sticks-based threshold due to insignificant correlation to the exposure lab threshold for n-butanol in our analyses.

Reply: The way of randomization is now described as follows: …
“Participants were assigned according to an a-priori computed randomization scheme to one of two groups which differed in the order the following detection tests were presented (see Figure 1): Half of the participants (group 1) first completed the olfactometer threshold assessment in groups of two participants, answered the Chemical Sensitivity Scale for Sensory Hyperreactivity [16] and the trait version of the Positive and Negative Affect Schedule [17]. The other half of the participants (group 2) first were administered individually the Sniffin’ Sticks threshold test and a lung function test (VitaloGraph, Hamburg, Germany).”

Round 2

Reviewer 1 Report

I believe the paper is improved and is now suitable for pubblication

Reviewer 2 Report

Many thanks to the authors for the improvement of the method description and the modification of the manuscript. If the experimental results can be compared more systematically, readers will be able to understand the difference in sensitivity of the method.
For example, what kind of systematic error is the exchange of  participant blocks  in the experimental procedure of Figure 1? The presentation in Figure 3 is difficult to distinguish the topics that you want to discuss.

Reviewer 3 Report

the paper is now acceptable for publication.